# CXCL13/CXCR5 Interaction Facilitates VCAM-1-Dependent Migration in Human Osteosarcoma

**DOI:** 10.3390/ijms21176095

**Published:** 2020-08-24

**Authors:** Ju-Fang Liu, Chiang-Wen Lee, Chih-Yang Lin, Chia-Chia Chao, Tsung-Ming Chang, Chien-Kuo Han, Yuan-Li Huang, Yi-Chin Fong, Chih-Hsin Tang

**Affiliations:** 1School of Oral Hygiene, College of Oral Medicine, Taipei Medical University, Taipei City 11031, Taiwan; jufangliu@tmu.edu.tw; 2Department of Orthopaedic Surgery, Chang Gung Memorial Hospital, Puzi City, Chiayi County 61363, Taiwan; cwlee@mail.cgust.edu.tw; 3Department of Nursing, Division of Basic Medical Sciences, and Chronic Diseases and Health Promotion Research Center, Chang Gung University of Science and Technology, Puzi City, Chiayi County 61363, Taiwan; 4Research Center for Industry of Human Ecology and Research Center for Chinese Herbal Medicine, Chang Gung University of Science and Technology, Guishan Dist., Taoyuan City 33303, Taiwan; 5School of Medicine, China Medical University, Taichung 40402, Taiwan; p123400@hotmail.com; 6Department of Respiratory Therapy, Fu Jen Catholic University, New Taipei City 24205, Taiwan; 095457@mail.fju.edu.tw; 7School of Medicine, Institute of Physiology, National Yang-Ming University, Taipei City 11221, Taiwan; briancoinage@ym.edu.tw; 8Department of Biotechnology, College of Health Science, Asia University, Taichung 40402, Taiwan; jackhan@asia.edu.tw (C.-K.H.); yuanli@asia.edu.tw (Y.-L.H.); 9Department of Sports Medicine, College of Health Care, China Medical University, Taichung 40402, Taiwan; 10Department of Orthopedic Surgery, China Medical University Beigang Hospital, Yunlin 65152, Taiwan; 11Department of Pharmacology, School of Medicine, China Medical University, Taichung 40402, Taiwan; 12Chinese Medicine Research Center, China Medical University, Taichung 40402, Taiwan

**Keywords:** CXCL13, CXCR5, migration, invasion, osteosarcoma

## Abstract

Osteosarcoma is the most common primary tumor of the skeletal system and is well-known to have an aggressive clinical outcome and high metastatic potential. The chemokine (C-X-C motif) ligand 13 (CXCL13) plays a vital role in the development of several cancers. However, the effect of CXCL13 in the motility of osteosarcoma cells remains uncertain. Here, we found that CXCL13 increases the migration and invasion potential of three osteosarcoma cell lines. In addition, CXCL13 expression was upregulated in migration-prone MG-63 cells. Vascular cell adhesion molecule 1 (VCAM-1) siRNA and antibody demonstrated that CXCL13 promotes migration via increasing VCAM-1 production. We also show that CXCR5 receptor controls CXCL13-mediated VCAM-1 expression and cell migration. Our study identified that CXCL13/CXCR5 axis facilitate VCAM-1 production and cell migration in human osteosarcoma via the phospholipase C beta (PLCβ), protein kinase C α (PKCα), c-Src, and nuclear factor-κB (NF-κB) signaling pathways. CXCL13 and CXCR5 appear to be a novel therapeutic target in metastatic osteosarcoma.

## 1. Introduction

Osteosarcoma is the most common primary bone tumor and is highly malignant [1]. Neoadjuvant and adjuvant chemotherapy for osteosarcoma has largely remained unchanged since the 1980s, with wide surgical excision and preservation of limb function [1]. However, despite such treatment, only 60–70% of patients without clinically evident metastasis at first diagnosis remain alive after 3 years [2]. Major challenges that reduce overall survival of patients with osteosarcoma include the development of chemoresistance and progression to lung metastasis [3,4,5]. Since the lung is the most frequent organ affected by metastasis in osteosarcoma [2], the development of therapeutic strategies for delaying or inhibiting lung metastasis progression remains critical for improving patient survival outcomes.

The metastatic process is a multi-stage cascade, whereby cancer characterized by the dissemination of tumor cells with proteolytic activity, enabling the tumor to invade various tissues [1]. Several different adhesion molecules regulate cancer migration and invasion during the metastatic process, including vascular cell adhesion molecule 1 (VCAM-1), an inducible surface glycoprotein from the immunoglobulin superfamily, which is involved in numerous biological activities [6]. VCAM-1 is highly expressed in metastatic cancer cells [7]. In osteosarcoma, levels of VCAM-1 expression correlate with disease stage and tumor progression [4], with higher levels of VCAM-1 expression enabling connective tissue growth factor to stimulate the migration of osteosarcoma cells and promote metastasis [8]. Thus, inhibiting VCAM-1 expression may be an effective strategy for treating osteosarcoma metastasis.

Chemokines are critical players in various cellular processes, contributing to the homing, migratory, angiogenic, and proliferative activities of cells [9]. They also regulate various cancer functions including progression, metastasis, apoptosis, and chemoresistance [10,11,12]. The chemokine (C-X-C motif) ligand 13 (CXCL13) plays a vital role in the development of several cancers (e.g., gastric, colorectal, breast and lung cancers), the promotion of angiogenesis, and the metastatic cascade [13,14,15,16]. Elevated levels of CXCL13 and its cognate receptor, CXCR5, are found in many types of cancer cells [16]. However, little is known about the CXCL13/CXCR5 signaling axis in osteosarcoma. Here, we report that the CXCL13/CXCR5 interaction promotes the migratory and invasive potential of human osteosarcoma cells, which in turn mediates the phospholipase C β (PLCβ), protein kinase C α (PKCα), c-Src, and nuclear factor-κB (NF-κB) pathways.

## 2. Results

### 2.1. CXCL13 Promotes Migration and Invasion of Human Osteosarcoma Cells

CXCL13 promotes the migration and metastases of cancer cells [16]. We first applied CXCL13 to human osteosarcoma cell lines and examined cell motility. Treatment of human osteosarcoma cell lines (MG-63, HOS, and U2OS) with CXCL13 (3–30 ng/mL) concentration-dependently facilitated migratory and invasive abilities (Figure 1A,B), but did not affect the viability of osteosarcoma cells (Figure 1C). Next, we investigated whether CXCL13 expression influences the migratory activity of osteosarcoma tumor cells. We previously established high-migration-prone sublines MG-63 (M2), MG-63 (M5), and MG-63 (M10) [17]. Our results in this study were similar to those in our previous experiment, which showed that migration-prone sublines MG-63 (M10) and MG-63 (M5) had higher cell motility than the original MG-63 cell line (Figure 1D). We also found that CXCL13 levels were upregulated in the high-migration-prone sublines (Figure 1E,F). Transfection of MG-63 (M10) cells with CXCL13 siRNA inhibited CXCL13 expression and cell migratory activity (Figure 1G–I). Transfecting MG-63 cells with CXCL13 siRNA did not significantly affect cell proliferation (Appendix A). Thus, these results indicate that CXCL13 promotes human osteosarcoma cell migration.

### 2.2. CXCL13 Promotes Osteosarcoma Cell Migration by Increasing VCAM-1 Expression

VCAM-1-dependent motility is essential for the development of tumor metastases [16,18]. We wondered if VCAM-1 also mediates the effects of CXCL13 upon cell migration. Incubation of osteosarcoma cells with CXCL13 increased VCAM-1 expression (Figure 2A). Transfection of cells with VCAM-1 siRNA reduced levels of VCAM-1 expression and CXCL13-induced promotion of osteosarcoma cell migration (Figure 2B–D), indicating that CXCL13 facilitates VCAM-1-dependent osteosarcoma cell migration.

### 2.3. The CXCL13/CXCR5 Axis Stimulates Osteosarcoma Cell Migration by Increasing VCAM-1 Expression

CXCR5 is a specific CXCL13 receptor that mediates CXCL13-regulated cancer functions [19]. Here, we found that transfecting cells with CXCR5 siRNA reversed the effects of CXCL13 upon cell migration and VCAM-1 expression (Figure 3A–C). Similarly, the CXCR5 receptor antibody reduced CXCL13-induced increases in cell migration and VCAM-1 production (Figure 3D–F). Therefore, the CXCL13/CXCR5 interaction regulates VCAM-1 synthesis and osteosarcoma cell migration. A comparison of levels of CXCR5 protein expression in all three osteosarcoma cell lines demonstrated that CXCR5 expression correlates with malignancy (Appendix A).

### 2.4. The PLCβ, PKCα, c-Src, and NF-κB Pathways Are Involved in CXCL13-Induced VCAM-1 Expression and Cell Migration

The PLCβ, PKCα, and c-Src signaling pathways are commonly identified during cancer metastasis [20,21]. Incubation of all three osteosarcoma cell lines with PLCβ, PKCα, and c-Src inhibitors (U73122, GF109203X, and PP2, respectively) antagonized CXCL13-induced increases in cell migration and VCAM-1 production (Figure 4A–C and Appendix A). Likewise, PLCβ, PKCα, and c-Src siRNAs reduced CXCL13-induced stimulation of osteosarcoma cell migration D and Appendix A). Stimulation of the cells with CXCL13 promoted PLCβ, PKCα, and c-Src phosphorylation in a time-dependent manner (Figure 4E). These data suggest that CXCL13 promotes VCAM-1-dependent migration via the PLCβ, PKCα, and c-Src pathways.

NF-κB is an important transcriptional factor that mediates cancer cell migration and metastasis [22,23]. Treating cells with NF-κB inhibitors (BAY 11-7082, PDTC, and TPCK) inhibited CXCL13-induced increases in cell migration and VCAM-1 expression (Figure 5A–C and Appendix A). Transfection of cells with p65 siRNA also reduced CXCL13-induced increases in cell migration and VCAM-1 production (Figure 5D–F and Appendix A). Treatment of osteosarcoma cells with CXCL13 facilitated IKK, IκBα, and p65 phosphorylation in a time-dependent manner (Figure 5G).

We then used the NF-κB luciferase promoter plasmid to determine the possibility that the PLCβ, PKCα, and c-Src signaling pathways mediate the effects of CXCL13 upon NF-κB activation. Our results revealed that NF-κB luciferase activity was augmented by CXCL13 and this effect was prevented when we pretreated the cells with PLCβ, PKCα, and c-Src inhibitors. (Figure 6A). The chromatin immunoprecipitation (ChIP) assay also found that these pathway inhibitors reduced CXCL13-induced binding of p65 to the NF-κB binding site on the VCAM-1 promotor (Figure 6B). Activation of PLCβ, PKCα, and c-Src appears to be necessary for CXCL13-induced NF-κB activation in human osteosarcoma cells.

## 3. Discussion

Osteosarcoma is the most common primary tumor of the skeletal system and is well-known to have an aggressive clinical outcome and high metastatic potential [24]. The high propensity for this disease to develop chemoresistance and progression to lung metastasis means that it is critical to develop an effective adjuvant therapy for inhibiting osteosarcoma metastasis. Chemokines are upregulated in several cancer types and can facilitate cancer cell proliferation, control apoptosis, survival, and metastasis. Although CXCL13-regulated cancer cell migration and metastasis has been reported in some cancers [16], the effect of CXCL13 in the motility of osteosarcoma cells remains uncertain. Numerous studies have shown that human physiological concentrations of CXCL13 are significantly increased in cancer patients compared with healthy controls [25]. Other cancer researchers have used treatment concentrations of CXCL13 in the range of 50–100 ng/mL [26,27], so the concentrations used in our study are physiologically relevant. Here, we found that CXCL13 increases the migration and invasion potential of three osteosarcoma cell lines. Interestingly, CXCL13 expression was upregulated in migration-prone MG-63 cells, indicating that CXCL13 levels are associated with the migratory ability of osteosarcoma cells. Treatment of MG-63 cells with inhibitors or transfection with their respective siRNAs did not significantly impact cell proliferation or migration as well as basal CXCL13 and CXCR5 expression, which suggests that these treatments did not affect cell viability (Appendix A). Moreover, we found that CXCL13 facilitates VCAM-1-dependent motility via the PLCβ, PKCα, c-Src, and NF-κB signaling cascades.

Chemokines and their receptors are capable of modulating the proliferative, angiogenic, migratory and invasive activities of cancer cells, so are critical to the progression and metastasis of cancers [28]. CXCL12 and its cell surface receptor CXCR4 combine to enhance the migration of breast cancer cells [29]. The CCL5/CCR5 interaction appears to modulate directional tumor migration and metastasis in different types of cancers, such as lung and breast cancer, as well as osteosarcoma [30,31,32]. Here, we found that CXCR5 siRNA and CXCR5 antibody markedly inhibited CXCL13-induced VCAM-1 expression and cell migration, suggesting that the CXCL13/CXCR5 interaction regulates VCAM-1-mediated migration of human osteosarcoma cells.

Many cellular functions are regulated by the activation of the PLCβ, PKCα, and c-Src signaling pathways [20,33], which also control adhesion molecule production and cell motility [21,34]. In this study, we found that CXCL13 facilitates PLCβ, PKCβ, and c-Src phosphorylation, while PLCβ, PKCα, and c-Src inhibitors reduce CXCL13-induced stimulation of VCAM-1 production and osteosarcoma cell migration. PLCβ, PKCα, and c-Src siRNAs confirmed the reversal of CXCL13-induced promotion of osteosarcoma cell migratory activity. The PLCβ, PKCα, and c-Src signaling pathways appear to play a role in CXCL13-facilitated VCAM-1-dependent osteosarcoma cell migration.

Several transcription factor-binding sites have been identified in the 5-regulatory region [4,35]. NF-κB is a vital transcriptional factor that mediates *VCAM-1* transcriptional activity and cancer metastasis [35]. The results of our study show that all three NF-κB inhibitors (BAY 11-7082, PDTC, and TPCK) antagonized the stimulatory effects of CXCL13 upon VCAM-1 expression and cellular migratory activity, indicating the importance of NF-κB activation in these processes. Our data also show that CXCL13 facilitates p65 binding to the NF-κB binding site on the VCAM-1 promotor and NF-κB luciferase activity. The antagonistic effects of PLCβ, PKCα, and c-Src inhibitors upon CXCL13-mediated activities suggest that these are regulated by the PLCβ, PKCα, and c-Src pathways.

## 4. Materials and Methods

### 4.1. Materials

We purchased p-PLCβ, PLCβ, p-PKCα, PKCα, p-c-Src, c-Src, p-p65, p65, CXCL13, VCAM-1, p-IKK, IKK, p-IκBα, IκBα, and actin from GeneTex International Corporation (Hsinchu City, Taiwan). All ON-TARGET*plus* siRNAs were purchased from Dharmacon (Lafayette, CO, USA). CXCR5 mAb was purchased from R&D Systems (Minneapolis, MN, USA). Cell culture supplements were purchased from Invitrogen (Carlsbad, CA, USA). A Dual-Luciferase^®^ Reporter Assay System was bought from Promega (Madison, WI, USA). qPCR primers and probes, as well as the Taqman^®^ One-Step PCR Master Mix, were supplied by Applied Biosystems (Foster City, CA, USA). The PLCβ inhibitor (U73122), PKCα inhibitor (GF109203X), c-Src inhibitor (PP2), IκB inhibitor (TPCK), NF-κB inhibitor (PDTC), NF-κB inhibitor (BAY 11-7082), and other chemicals not already mentioned were supplied by Sigma-Aldrich (St. Louis, MO, USA).

### 4.2. Cell Culture

The human osteosarcoma cell lines (MG-63, HOS, and U2OS) were obtained from the American Type Cell Culture Collection (Manassas, VA, USA). MG-63 and HOS cells were cultured in MEM medium and U2OS cells were cultured in McCoy’s 5A medium. All cultured mediums were supplemented with 10% FBS and antibiotics then maintained in a humidified incubator at 37 °C in 5% CO_2_.

The migration-prone MG-63(M2), MG-63(M5), and MG-63(M10) cells were established from original MG-63 cells, as according to the methods described in our previous reports [4,17].

### 4.3. Western Blot Analysis

We applied the SDS-PAGE procedure to resolve extracted proteins, before transferring them to Immobilon^®^ PVDF membranes. Analysis of the proteins was performed by Western blot, following the procedure detailed in our previous publications [36,37,38].

### 4.4. mRNA Quantification

Total RNA was extracted from osteosarcoma cells using TRIzol reagent. Quantitative real-time PCR (qPCR) analysis was conducted as according to our previous reports [39,40].

### 4.5. Migration and Invasion Assay

The migration and invasion assays were performed with Transwell plates (Costar, NY, USA), as according to our previous publications [17,41]. Migrated cells were fixed and stained with crystal violet, then manually counted under a microscope.

### 4.6. Luciferase Reporter Assay

Osteosarcoma cells were transfected for 24 h with κB-luciferase reporter gene construct, following the Lipofectamine^™^ 2000 Reagent protocol. The cells were then exposed to CXCL13 and the Promega luciferase activity kit measured luciferase activity [39,42,43].

### 4.7. Chromatin Immunoprecipitation (ChIP) Assay

DNA were extracted from osteosarcoma cells and immunoprecipitated using anti-p65 antibody. The SimpleChIP^®^ Enzymatic Chromatin IP kit (Cell Signaling, MA, USA) was used to conduct ChIP assays. The purified DNA pellet was subjected to PCR testing, as detailed in our previous publication [4].

### 4.8. Statistics

All values represent mean ± standard deviation (S.D.). The two-tailed Student’s t-test assessed the significance of between-group differences, using a cut-off point of 0.05.

## 5. Conclusions

Our study has identified that the CXCL13/CXCR5 axis facilitates VCAM-1 production and the migration of human osteosarcoma cells via the PLCβ, PKCα, c-Src, and NF-κB signaling pathways (Figure 7). CXCL13 and CXCR5 appear to be novel therapeutic targets in metastatic osteosarcoma.

## Figures and Tables

**Figure 1 ijms-21-06095-f001:**
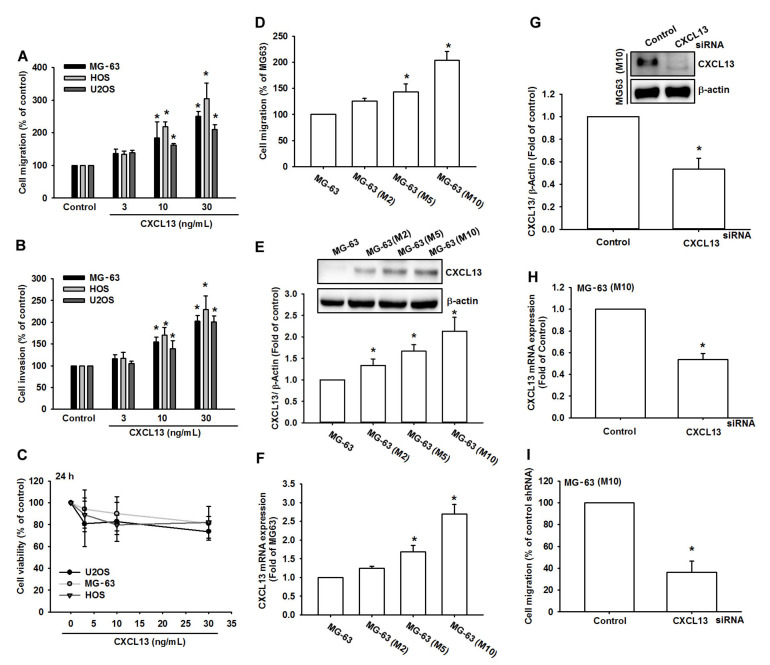
The chemokine (C-X-C motif) ligand 13 (CXCL13) increases migration and invasion in human osteosarcoma. (**A**,**B**) Cells were incubated with CXCL13 (3–30 ng/mL) and the Transwell assay determined in vitro migratory and invasion activity after 24 h. (**C**) Cells were incubated with CXCL13 for 24 h and cell viability was examined by the MTT assay. (**D**–**F**) Migratory ability and CXCL13 expression of the indicated cells was examined by Transwell, Western blot and qPCR assays. (**G**–**I**) Cells were transfected with CXCL13 siRNA then stimulated with CXCL13; CXCL13 expression and migratory potential was examined by Western blot, qPCR, and Transwell assays. * *p* < 0.05 compared with the control group.

**Figure 2 ijms-21-06095-f002:**
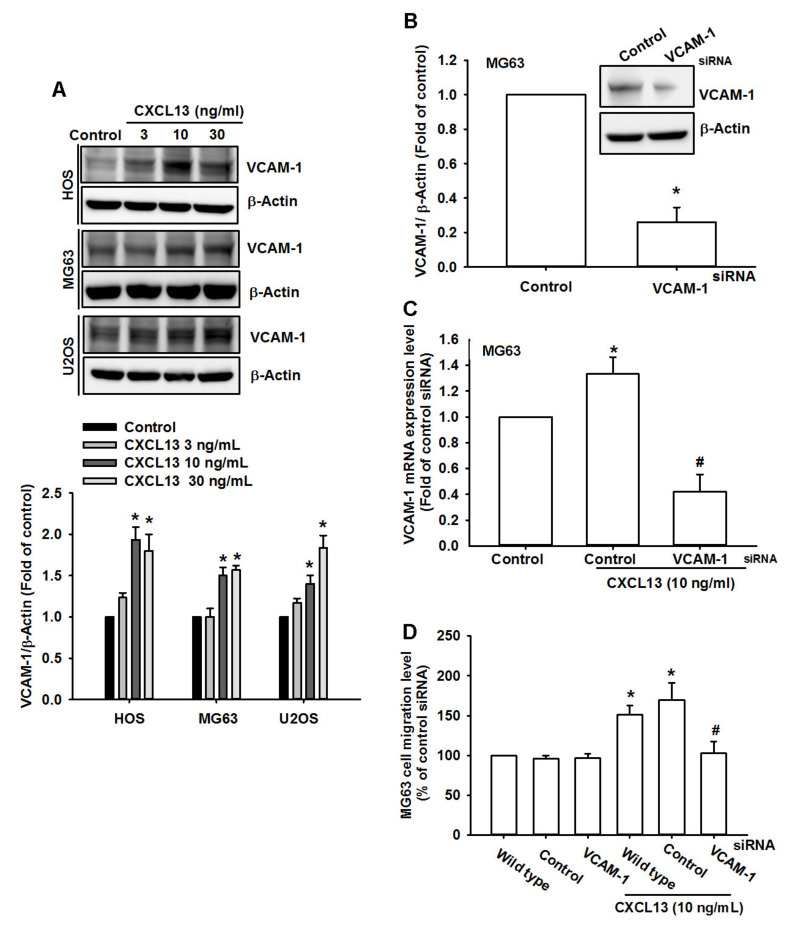
CXCL13 promotes osteosarcoma migration by increasing vascular cell adhesion molecule 1 (VCAM-1) expression. (**A**) Cells were incubated with CXCL13 (3–30 ng/mL) and VCAM-1 expression was examined by Western blot. (**B**–**D**) MG-63 cells were transfected with VCAM-1 siRNA then stimulated with CXCL13; VCAM-1 expression and migratory potential was examined by Western blot, qPCR, and Transwell assays. * *p* < 0.05 compared with the control group; # *p* < 0.05 compared with the CXCL13-treated group.

**Figure 3 ijms-21-06095-f003:**
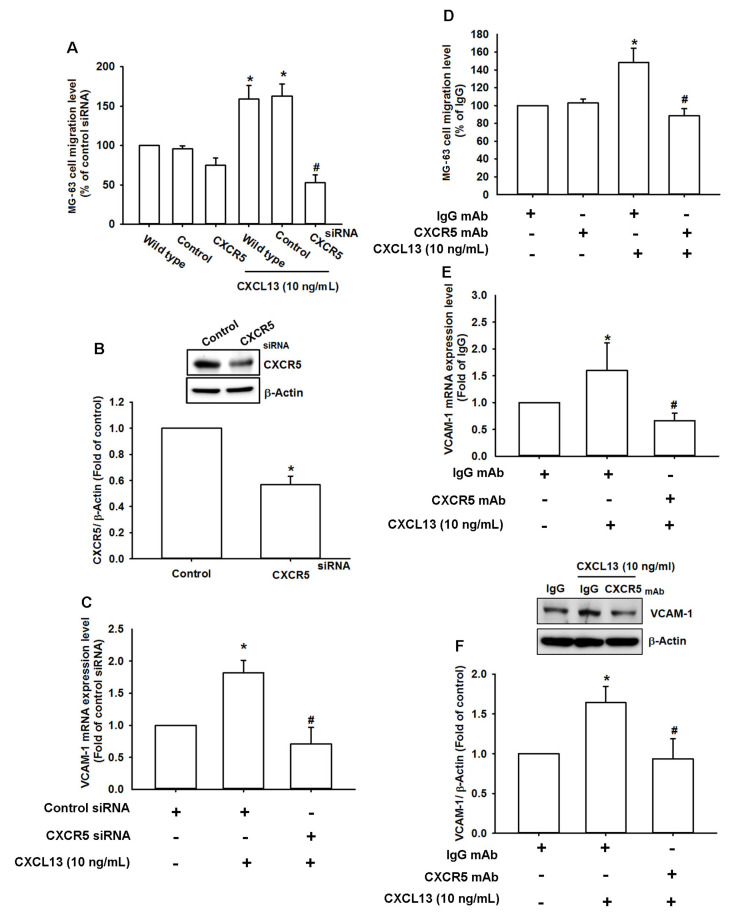
The CXCL13/CXCR5 axis increases VCAM-1 expression and cell migration. MG-63 cells were transfected with CXCR5 siRNA (**A**–**C**) or pretreated with CXCR5 antibody (**D**,**E**), then stimulated with CXCL13; migratory potential and VCAM-1 expression was examined by Transwell, qPCR and Western blot assays. (**F**) MG-63 cells were pretreated with CXCR5 antibody then stimulated with CXCL13; VCAM-1 protein expression was examined by Western blot assay. * *p* < 0.05 compared with the control group; # *p* < 0.05 compared with the CXCL13-treated group.

**Figure 4 ijms-21-06095-f004:**
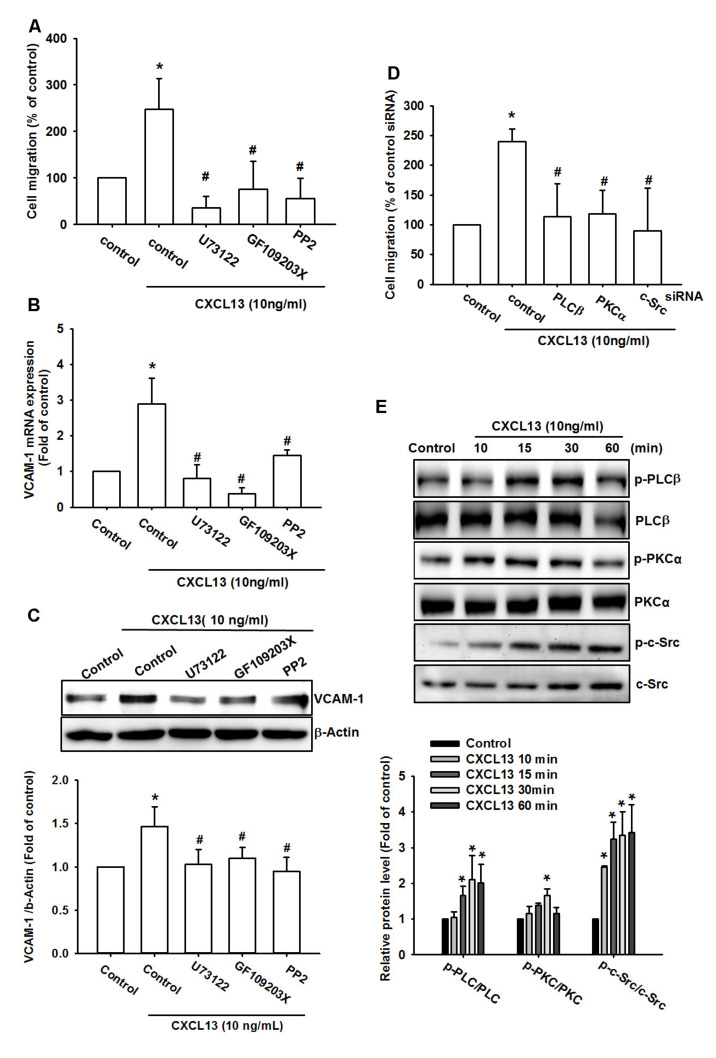
The phospholipase C β (PLCβ), protein kinase C α (PKCα), and c-Src pathways are involved in CXCL13-promoted migration and VCAM-1 production. (**A**–**C**) MG-63 cells were pretreated with U73122, GF109203X, and PP2 then stimulated with CXCL13; migratory potential and VCAM-1 expression was examined by Transwell, qPCR and Western blot assays. (**D**) MG-63 cells were transfected with PLCβ, PKCα, and c-Src siRNA then stimulated with CXCL13; migratory potential was examined by the Transwell assay. (**E**) MG-63 cells were incubated with CXCL13 for the indicated time intervals and PLCβ, PKCα and c-Src phosphorylation was examined by Western blot. * *p* < 0.05 compared with the control group; # *p* < 0.05 compared with the CXCL13-treated group.

**Figure 5 ijms-21-06095-f005:**
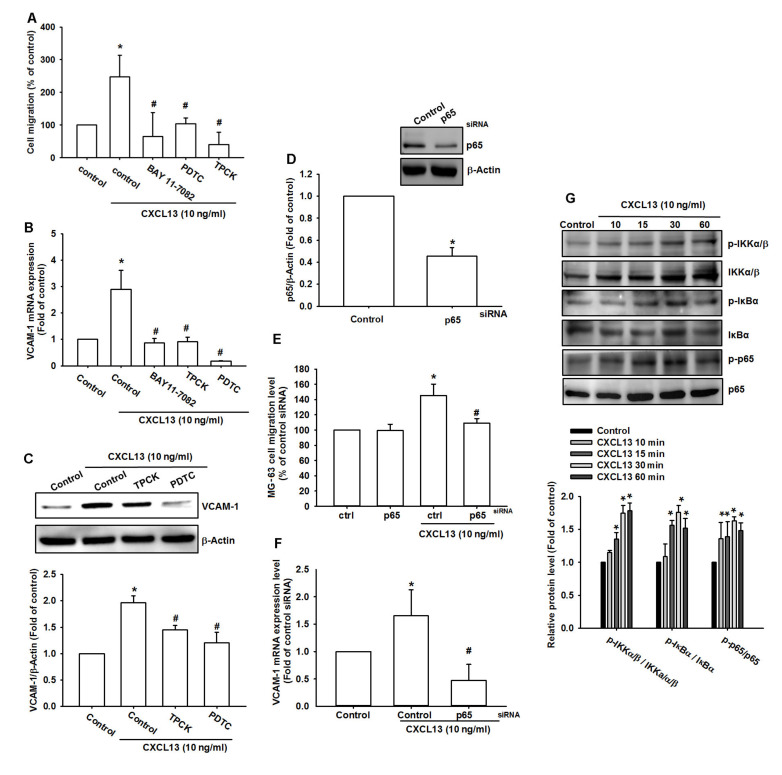
The transcription factor nuclear factor-κB (NF-κB) is involved in CXCL13-induced cell migration and VCAM-1 production. MG-63 cells were pretreated with BAY 11-7082, PDTC, or TPCK (**A**–**C**), or transfected with p65 siRNA (**D**–**F**), then stimulated with CXCL13; cell migratory potential and VCAM-1 expression was examined by Transwell, qPCR, and Western blot assays. (**G**) MG-63 cells were incubated with CXCL13 for the indicated time intervals, and IKK, IκBα, and p65 phosphorylation was examined by Western blot. * *p* < 0.05 compared with the control group; # *p* < 0.05 compared with the CXCL13-treated group.

**Figure 6 ijms-21-06095-f006:**
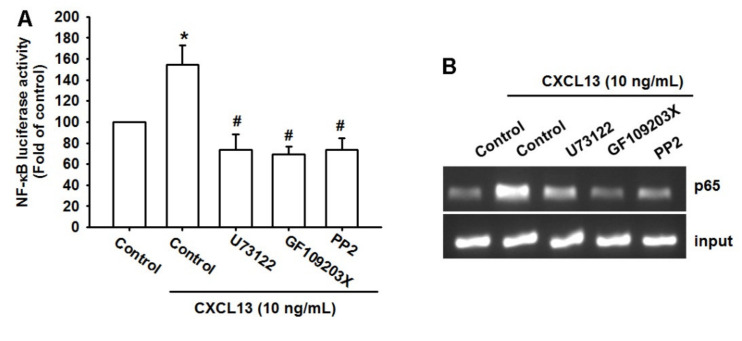
The PLCβ, PKCα, and c-Src pathways regulate CXCL13-induced NF-κB activation. MG-63 cells were pretreated with U73122, GF109203X, or PP2, then stimulated with CXCL13; NF-κB luciferase activity and p65 binding to the NF-κB site on the VCAM-1 promoter was examined using the luciferase (**A**) and chromatin immunoprecipitation (ChIP) assays (**B**). * *p* < 0.05 compared with the control group; # *p* < 0.05 compared with the CXCL13-treated group.

**Figure 7 ijms-21-06095-f007:**
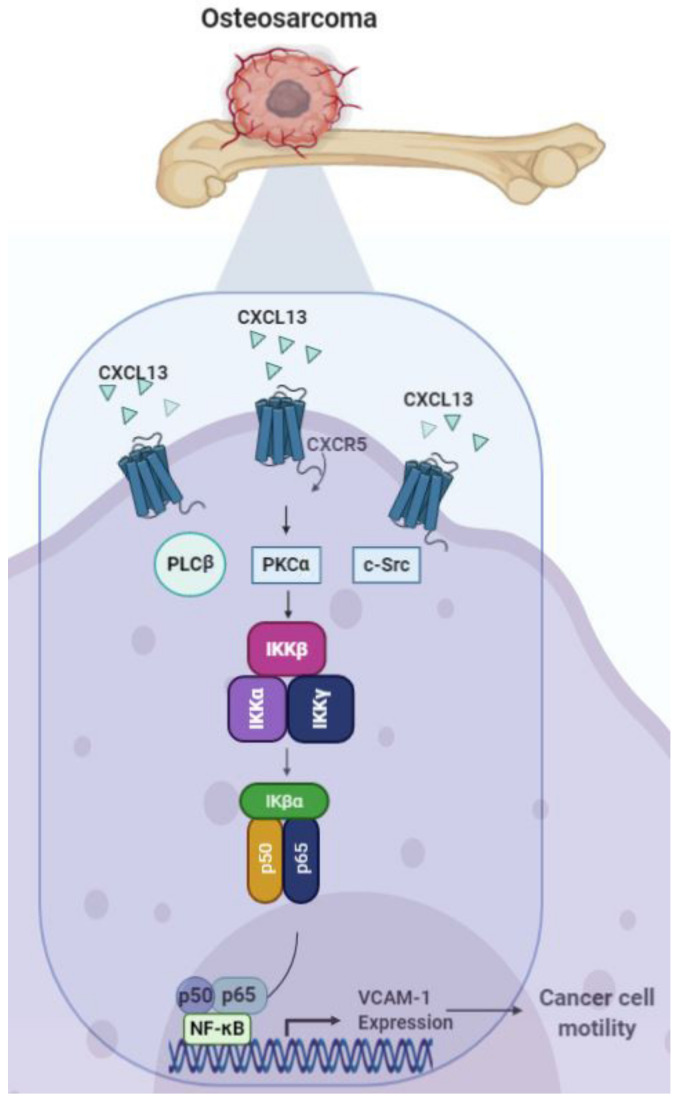
Schema illustrating the effects of the CXCL13/CXCR5 axis on osteosarcoma cell migration. The CXCL13/CXCR5 interaction facilitates VCAM-1-dependent migration through the PLCβ, PKCα, c-Src, and NF-κB signaling pathways in human osteosarcoma cells.

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
