# Peer review of "CXCL13/CXCR5 Interaction Facilitates VCAM-1-Dependent Migration in Human Osteosarcoma"

_ijms, 2020, doi:10.3390/ijms21176095_

Round 1
Reviewer 1 Report
The authors investigated the role of CXCL13/CXCR5 signaling axis in human osteosarcoma (OS) cells. They found that CXCL13 treatment promotes migratory and invasive abilities in three human OS cell lines (MG-63, HOS, U2OS). CXCL13/CXCR5 signaling axis was associated with activation of VCAM-1 and its downstream target genes in MG-63 cells. Although these findings are potentially valuable, there are many points that require clarification as follows:
Major issues:
- The authors demonstrated that CXCL13 treatment promotes migratory and invasive abilities in three human ESCC cell lines (MG-63, HOS, U2Os). However, only one MG-63 cells were used to confirm the underlying mechanism of CXCL13-mediated signaling pathway in human OS cells. The authors should use at least two human OS cells to investigate the role of CXCL13-mediated signaling pathway in this study.
- The authors showed that CXCL13 treatment increases the expression of VCAM-1 in three human OS cells (Fig. 2A). However, there was no upregulation of VCAM-1 in CXCL13-treated U2OS cells. Although there were two bands for VCAM-1, which band was VCAM-1? Please add the quantitative data for the expression level of VCAM-1 protein in Fig. 2A.
Minor issues:
- The authors should show the expression level of CXCR5 in three human OS cells.
- The authors should correct “CXCR13” to “CXCL13” in the result section (page 5) and discussion section (page 7).
Author Response
Reviewer 1
Q1. The authors demonstrated that CXCL13 treatment promotes migratory and invasive abilities in three human ESCC cell lines (MG-63, HOS, U2Os). However, only one MG-63 cells were used to confirm the underlying mechanism of CXCL13-mediated signaling pathway in human OS cells. The authors should use at least two human OS cells to investigate the role of CXCL13-mediated signaling pathway in this study.
Answer: We thank the reviewer for this suggestion and have provided the requested data in the Supplementary files (Fig. S3A-D). Pretreatment of HOS and U2OS cells with U73122, GF109203X, PP2, PDTC, or TPCK antagonized CXCR13-induced increases in cell migration (Fig. S3A-B). Likewise, PLCα, PKCβ and c-Src siRNAs all reduced CXCR13-induced stimulation of osteosarcoma cell migration (Fig. S3C-D).
Q2. The authors showed that CXCL13 treatment increases the expression of VCAM-1 in three human OS cells (Fig. 2A). However, there was no upregulation of VCAM-1 in CXCL13-treated U2OS cells. Although there were two bands for VCAM-1, which band was VCAM-1? Please add the quantitative data for the expression level of VCAM-1 protein in Fig. 2A.
Answer: We thank the reviewer for this feedback, which has enabled us to improve the data provided in Figure 2A, which now more clearly illustrates the findings from our Western blot assays and provides quantification results for VCAM-1 protein expression.
Q3. The authors should show the expression level of CXCR5 in three human OS cells.
Answer: We thank the reviewer for this comment. We now show levels of CXCR5 expression for all three human osteosarcoma cell lines (MG-63, HOS, U2OS) in Figure S2. The findings suggest that CXCR5 expression correlates with malignancy. U2OS and HOS cells showed obvious high CXCR5 expression (Fig. S2).
Q4. The authors should correct “CXCR13” to “CXCL13” in the result section (page 5) and discussion section (page 7).
Answer: We thank the reviewer for this careful attention to our manuscript. CXCL13 has been corrected in the revised version.
Reviewer 2 Report
Ju-Fang Liu et al., identified using osteosarcoma cell lines that cell migration is induced by the CXCL13/CXCR5 axis. CXCL13, binding its receptor on the cell membrane (CXCR5), induces VCAM-1 expression via the PLCα, PKCβ, c-Src and NK-κβ signaling pathways. Even if the study is interesting, several additional experiments are required before publication in order to support author’s conclusions.
- The authors should explain which are the human physiological concentrations of CXCL13, thus if the concentrations used in this study are physiological relevant;
- Lane 88 and Fig.1H, 1I: change shRNA with siRNA;
- WB experiments in Fig. 1E and Fig. 1G should be quantified and statistics should be reported;
- The reduced cell migration phenotype in CXCL13 KD condition may be due to a decrease in cell viability. Cell invasion experiments and MTT assay to check cell viability should be performed in CXCL13 siRNA condition (see Fig. 1G-I);
- WB experiments in Fig. 2A and 2B should be quantified and statistics should be reported;
- The authors should explain why in Fig. 2B decided to use 10 ng/ml of CXCL13 and not the maximum concentration used in Fig. 1 (30 ng/ml);
- Please clarify which sublines (MG63) was used in Fig. 2B and 2C.
- In Fig. 2C the condition “VCAM-1” is not clear. For me the groups that should be compared are: MG63 (no siRNA, minus CXCL13), MG63 (no siRNA, plus CXCL13), MG63 ( ctrl siRNA, minus CXCL13), MG63 (plus VCAM-1 siRNA, plus CXCL13). Please explain also in Fig. 3;
- Cell viability and cell invasion upon VCAM-1 KD should be checked;
- The expression level of CXCR5 upon siRNA KD should be quantified and the statistics reported in Fig. 3B. Moreover, using WB, the expression level of VCAM-1 should be analysed and quantified, same for Fig. 3D);
- WB experiment in Fig. 3E should be quantified and statistics should be reported;
- The mRNA and expression levels of CXCL13 upon CXCR5 KD should be checked;
- Cell viability and cell invasion upon CXCR5 KD and after the use of CXCR5 antibody application should be reported;
- In Fig. 4A, cell viability and cell invasion upon inhibitors treatment should be checked;
- CXCL13 and CXCR5 mRNA expression levels and protein expression should be checked and quantified upon inhibitors treatments in order to exclude possible off-target effects; Please check also using WB VCAM-1 expression;
- Please quantify the WB experiments in Fig. 4C and 4E; please also add statistics;
- In Fig. 5 cell viability and cell invasion upon inhibitor treatments should be reported;
- Also in Fig. 5: CXCL13 and CXCR5 mRNA expression levels and protein expression should be checked and quantified upon inhibitors treatments and p65 KD in order to exclude possible off-target effects; Please check also using WB VCAM-1 expression;
- Please quantify and report statistic of all WB in Fig. 5!
- Cell viability and cell invasion upon p65 KD should be analysed.
Author Response
Reviewers' comments:
Reviewer 2
Q1. The authors should explain which are the human physiological concentrations of CXCL13, thus if the concentrations used in this study are physiological relevant.
Answer: Numerous studies have shown that human physiological concentrations of CXCL13 are significantly increased in cancer patients compared with healthy controls [25]. Other cancer researchers have used treatment concentrations of CXCL13 in the range of 50–100 ng/ml [26, 27], so the concentrations used in our study are physiologically relevant. (Lines 169-172)
Q2. Lane 88 and Fig.1H, 1I: change shRNA with siRNA.
Answer: We thank the reviewer for this close attention to our manuscript and we have accordingly corrected the text in the revised manuscript.
Q3. WB experiments in Fig. 1E and Fig. 1G should be quantified and statistics should be reported.
Answer: We thank the reviewer for this feedback. Quantification results from Western blot assays have accordingly been provided in Figure 1E and 1G.
Q4. The reduced cell migration phenotype in CXCL13 KD condition may be due to a decrease in cell viability. Cell invasion experiments and MTT assay to check cell viability should be performed in CXCL13 siRNA condition (see Fig. 1G-I).
Answer: These results have been provided in Figure S1.
Q5. WB experiments in Fig. 2A and 2B should be quantified and statistics should be reported.
Answer: The quantification results from Western blot assays have been provided showing levels of VCAM-1 expression in response to different concentrations of CXCL13 in each of the three osteosarcoma cell lines, as well as a significant mean change in VCAM-1 expression after siRNA transfection.
Q6. The authors should explain why in Fig. 2B decided to use 10 ng/ml of CXCL13 and not the maximum concentration used in Fig. 1 (30 ng/ml).
Answer: Previous researchers have used concentrations of 50–100 mg/ml for CXCL13 treatment of cancer cell lines, so we initially used treatment concentrations raning from 3 to 30 ng/ml. After observing that there were no significant differences between the 10 ng/ml and 30 ng/ml doses in relation to in vitro migratory and invasion activity after 24 h (see Figure 1A&B), we realized that the higher dose was unnecessary for further experiments.
Q7. Please clarify which sublines (MG63) was used in Fig. 2B and 2C.
Answer: We thank the reviewer for this comment and we have accordingly amended the labeling of Figure 2B and 2C.
Q8. In Fig. 2C the condition “VCAM-1” is not clear. For me the groups that should be compared are: MG63 (no siRNA, minus CXCL13), MG63 (no siRNA, plus CXCL13), MG63 (ctrl siRNA, minus CXCL13), MG63 (plus VCAM-1 siRNA, plus CXCL13). Please explain also in Fig. 3.
We apologize for inadvertently introducing some confusion into the labels of Figures 2 and 3. The abbreviation of Ctrl in both figures represents the word “Control” – we have now amended the labeling in both figures to incorporate the full word “Control” and we have added more details to both figures. Figure 2C has been changed to Figure 2D.
Q9. Cell viability and cell invasion upon VCAM-1 KD should be checked.
Answer: These results have been provided in Figure S4B&D. VCAM-1 siRNA did not affect basal cell viability and cell migration ability.
Q10. The expression level of CXCR5 upon siRNA KD should be quantified and the statistics reported in Fig. 3B. Moreover, using WB, the expression level of VCAM-1 should be analysed and quantified, same for Fig. 3D).
Answer: All of these details have now been provided in Figure 3, which has been expanded to include more details.
Q11. WB experiment in Fig. 3E should be quantified and statistics should be reported.
Answer: These details are now provided.
Q12. The mRNA and expression levels of CXCL13 upon CXCR5 KD should be checked.
Answer: In this study, we purchased CXCR5 ON-TARGETplus siRNA from Dharmacon. This siRNA don’t have off-target effect. Therefore, don’t affect CXCL13 expression. We also found CXCR5 siRNA did not affect basal cell viability and cell migration ability Figure S4B&D.
Q13. Cell viability and cell invasion upon CXCR5 KD and after the use of CXCR5 antibody application should be reported.
Answer: These results have been provided in Figure S4, which shows that cell proliferation and migration were not significantly impacted by treatment with inhibitors or transfection with their respective siRNAs, which suggests that these treatments did not affect cell viability and cell migration ability (Suppl. Fig. 4).
Q14. In Fig. 4A, cell viability and cell invasion upon inhibitors treatment should be checked.
Answer: These results have been provided in Figure S4A&C.
Q15. CXCL13 and CXCR5 mRNA expression levels and protein expression should be checked and quantified upon inhibitors treatments in order to exclude possible off-target effects; Please check also using WB VCAM-1 expression.
Answer: In this study, the working concentration of pharmacological inhibitors were used according datasheet from Sigma-Aldrich (St. Louis, MO, USA), which don’t have off-target effect. We also found these inhibitors did not affect basal cell viability and cell migration ability Figure S4A&C.
Q16. Please quantify the WB experiments in Fig. 4C and 4E; please also add statistics.
Answer: These details have now been provided.
Q17. In Fig. 5 cell viability and cell invasion upon inhibitor treatments should be reported.
Answer: These results have been provided in Figure S4A&C.
Q18. Also in Fig. 5: CXCL13 and CXCR5 mRNA expression levels and protein expression should be checked and quantified upon inhibitors treatments and p65 KD in order to exclude possible off-target effects; Please check also using WB VCAM-1 expression.
Answer: In this study, we purchased all ON-TARGETplus siRNAs from Dharmacon. These siRNAs don’t have off-target effect. We also found these siRNAs did not affect basal cell viability and cell migration ability Figure S4B&D.
Q19. Please quantify and report statistic of all WB in Fig. 5!
Answer: These details have now been provided in Figure 5.
Q20. Cell viability and cell invasion upon p65 KD should be analysed.
Answer: These results have been provided in Figure S4 B&D.
Round 2
Reviewer 1 Report
The authors addressed the appropriate comments by showing the additional data in the revised Figures.
Author Response
We thank the reviewer for this feedback.
Reviewer 2 Report
The authors have addressed most of my concerns. However, few points still need to be addressed in order to fully support their publication:
- In Fig.S1 and S4 statistics should be reported.
- Even if the siRNA CXCR5 and p65 don’t have off-target effect (based on Dharmacon’s datasheet), the expression level of CXCL13 and CXCR5 (in p65 KD condition) should be checked. Indeed, in the first case, a reduction of the receptor may lead also to a decreased expression level of the ligand that authors could not exclude a priori.
- Cell viability and cell invasion experiments upon CXCR5 antibody are still missing.
- Please add in the material and methods section detailed information regarding the experiments where pharmacological inhibitors were used. Moreover, if the inhibitors where used for more than 12 hours, please check the levels of CXCL13 and CXCR5. The authors could not exclude an effect in CXCL13 and CXCR5 based on Sigma's datasheet .
Author Response
Q1- In Fig.S1 and S4 statistics should be reported.
Answer: We thank the reviewer for your comments. The statistics describe have been provided in supplementary data (Figure S1 and S4).
Q2-- Even if the siRNA CXCR5 and p65 don’t have off-target effect (based on Dharmacon’s datasheet), the expression level of CXCL13 and CXCR5 (in p65 KD condition) should be checked. Indeed, in the first case, a reduction of the receptor may lead also to a decreased expression level of the ligand that authors could not exclude a priori.
Answer: We thank the reviewer for your comments. The data have been provided in supplementary data (Figure S4F and H). p65 siRNA did not affect CXCL13 and CXCR5 expression. CXCR5 siRNA did not affect CXCL13 expression.
Q3- Cell viability and cell invasion experiments upon CXCR5 antibody are still missing.
Answer: We thank the reviewer for your comments. The data have been provided in supplementary data (Figure S4A and C).
Q4- Please add in the material and methods section detailed information regarding the experiments where pharmacological inhibitors were used. Moreover, if the inhibitors where used for more than 12 hours, please check the levels of CXCL13 and CXCR5. The authors could not exclude an effect in CXCL13 and CXCR5 based on Sigma's datasheet.
Answer: (i) We thank the reviewer for your comments. The detail information of pharmacology inhibitors has been provided in Method section. (ii) These results have been provided in Figure S4E&G.
Round 3
Reviewer 2 Report
The authors fully adressed my comments. I would like to suggest the publication